# The coevolution of contagion and behavior with increasing and decreasing awareness

**Samira Maghool[1¤], Nahid Maleki-Jirsaraei[1], Marco Cremonini[ID][2]***

**1** Complex Systems Laboratory, Physics Department, Alzahra University, Tehran, Iran, **2** Department of Social and Political Sciences, University of Milan, Milan, Italy

¤ Current address: Computer Science Department, University of Milan, Milan, Italy
* marco.cremonini@unimi.it

**Data Availability Statement:** We have set up a GitHub repository (https://github.com/mc-unimi/epidemics-awareness-paper) that contains both the code of the multiagent epidemic simulator (folder code) and the data as CSV files (folder

## Abstract

Understanding the effects of individual awareness on epidemic phenomena is important to comprehend the coevolving system dynamic, to improve forecasting, and to better evaluate the outcome of possible interventions. In previous models of epidemics on social networks, individual awareness has often been approximated as a generic personal trait that depends on social reinforcement, and used to introduce variability in state transition probabilities. A novelty of this work is to assume that individual awareness is a function of several contributing factors pooled together, different by nature and dynamics, and to study it for different epidemic categories. This way, our model still has awareness as the core attribute that may change state transition probabilities. Another contribution is to study positive and negative variations of awareness, in a contagion-behavior model. Imitation is the key mechanism that we model for manipulating awareness, under different network settings and assumptions, in particular regarding the degree of intentionality that individuals may exhibit in spreading an epidemic. Three epidemic categories are considered—disease, addiction, and rumor—to discuss different imitation mechanisms and degree of intentionality. We assume a population with a heterogeneous distribution of awareness and different response mechanisms to information gathered from the network. With simulations, we show the interplay between population and awareness factors producing a distribution of state transition probabilities and analyze how different network and epidemic configurations modify transmission patterns.

## Introduction

Epidemics on networks is a research topic that has been investigated for a long time. First developed to apply mathematical and statistical methods to the study of the spread of diseases in populations [1–3], epidemic models evolved to describe other viral phenomena unrelated to pathogens and often characterized by a behavioral or social dimension [4–6]. The spread of addictions, like heroin diffusion in the '70s [7], as well as the recent opioid addiction [8], has been studied with epidemic models, conveniently adapted from original disease-based ones. Also the diffusion of ideas [9] and of rumors [10] have been analyzed since the '60s by adapting epidemic models. The modern development of online social networks and phenomena like the

data). For the code we included the GPLv3 license; data is licensed under the Creative Commons Attribution 3.0 Unported license. A v1.0 release has been created and synchronized with Zenodo. The DOI is 10.5281/zenodo.3462521 (http://doi.org/10.5281/zenodo.3462521).

**Funding:** This work was partly supported by the EU-funded project THREAT-ARREST (contract n. H2020-786890). The authors received no additional funding for this work.

**Competing interests:** The authors have declared that no competing interests exist.

spread of misinformation and disinformation represent widely researched topics [11–15]. Other important extensions to original epidemic models have been the inclusion of adaptive agents, which allows for richer dynamics, multi-agent models [16] based on local rules that introduce self-organizing behaviors [17], or models with game-theoretic agents that simulate strategic behaviors [18].

The rich research strand that grew upon epidemic models has demonstrated the wide applicability and adaptability of original *SIS*, *SIR*, and *SIRS* models (i.e., where states stand for Susceptible, Infected, and Recovered), sometimes further extended with new specialized states [19–21].

With this work, we describe a multiagent model for the coevolving dynamics between an epidemic and agents' awareness, with the epidemic's dynamics influenced by the behavior of agents becoming aware of the epidemics. Similar coevolving dynamics are the subject of a rich strand of research often identified as *epidemic/behavior models* [21–25]. In these models, awareness has typically the role of the body of knowledge an agent acquires from the spread of an epidemic (e.g., as social reinforcement produced by multiple and repeated observations of peers, interactions, and communications, or from broadcasting media outlets and public bodies' initiatives) [26–28], and it is used as an attribute of infection and recovery probabilities, making them dependent to the behavioral response of agents [22, 23, 25, 27, 29]. With the extension of epidemic/behavior models to multiplex networks [30, 31], the coevolution between the epidemic spreading and agents' awareness has assumed the form of two network processes with mutual influence. In [32], the epidemic is modeled as a SIS process and awareness as a similar SIS-like dynamics indicated as *UAU*, with agents possibly cycling around *Unaware* and *Aware* status. Despite the work does not specifically investigate the case of agents reverting to unaware, being more focused on the possible role of a broadcasting media, the implicit non-monotonic dynamics of awareness is of particular interest for us. Another, more recent work also studied a UAU-SIS multiplex model [33], but the transition from Aware to Unaware state is only motivated with a memory loss effect. Several other studies have considered declining awareness as the result of memory loss, especially in absence of reinforcement events, like new outbreaks or awareness campaigns [25, 34, 35]. Although we recognize memory loss as an important effect and account for it in our definition of awareness as pooled factors, it is especially relevant in the long run, and it is a different scenario with respect to the one we have studied. Differently from all works considering only increasing awareness or explaining awareness decline with memory loss, in our work awareness could both increase and decrease during the whole dynamics, from the exponential start up growth to the regime reached after the initial peak. To model positive or negative awareness variations, we only introduce typical behavioral responses of agents to social reinforcement conditions, without any memory loss long-term assumption. In particular, negative changes to awareness are likely to be relevant and frequent in *non-disease* epidemics, because for addiction or rumor based epidemics, the common assumption of disease/behavior models stating that an individual tends to become more cautious being aware of infected neighbors [27, 35] should be questioned as not as prevalent as for diseases. In disease/behavior studies, the diffusion of anti-vaccination sentiments has often been framed as a social contagion resulting in increased vulnerability to outbreaks [36]. However, it represents a different case from what it is common in addictions and rumor spreading, where individuals, often as a response to social reinforcement, actively seek to develop an addiction or to believe and spread rumors. At best of our knowledge, ours is the first multiagent model explicitly considering positive and negative awareness variations as a behavioral response of an heterogeneous population to opposite social reinforcements. This generalization of the coevolution between a contagion process and agents' behavior, apparently more relevant for non-disease epidemics, also seems to justify the

introduction of a more general definition for this class of models as *contagion/behavior* models, rather than the more specific disease/behavior.

The reminder of the paper is organized as follows. The second section introduces our model, with the definition of awareness' components and the relation between state transition probabilities and agent awareness. The simulation results are then presented and discussed, organized for different epidemic categories and integrated with the specification of different characteristics of the imitation mechanism. A conclusion is presented in the last section.

## Awareness, imitation, and contagion dynamics

### Awareness contributing factors

Awareness, as a concept, has roots in psychology studies [37] and has often been associated to *metacognition* to describe self-reflection and understanding in learning [38], or specified as *situation awareness* as key for human decision making in dynamical systems [39]. Individual abilities and environmental factors have typically been investigated in awareness studies. All these elements, learning and decision making, personal traits and the context, are important for our work because they informed several design decisions for our contagion-behavior multiagent model. When awareness has been considered in coevolving epidemic models, it was often regarded as a measure of knowledge of some sort. Information about the existence of an epidemic has been associated to awareness, built upon the direct observation of infected neighbors or information received through means of communication [23, 25, 26, 28, 29].

A contribution of this work is to suggest that awareness should be defined as the combination, possibly variable and different for different type of agents, of multiple information sources. Information sources should not just vary in numbers, but also in nature, and it is this combined effect of heterogeneous factors that gives to awareness its characteristic feature of being partially situational awareness, i.e. "adaptive, externally directed consciousness" [40], and partially the result of personal traits, culture, education, and knowledge, in some sense similar to the assumption of imperfect knowledge and bounded rationality in risk prospects [41].

Following this logic, with this work we have started by considering three information sources as contributing to awareness: *Self-awareness*, *Imitation*, and *Communication*.

**Self-awareness.** It represents the awareness an individual has before the beginning of an epidemic and derived from her/his personal body of knowledge. It could depend from education, expertise, or personal skills. We use this factor to set up a heterogeneous population of agents starting with different level of awareness, and different responses to the contagion-behavior dynamics. We defined three agent types based on the value and usage of this factor:

- No Awareness;

- Low Self-awareness;

- High Self-awareness.

*No Awareness* type represents those whose dynamics is a simple contagion using fixed state transition probabilities and serves the purpose of benchmark for the other two types. *Low* and *High Self Awareness* agents differs for the initial level of self-awareness and represent the two agent having different behavioral responses to social reinforcement. Their state transition probabilities are variable with respect to their awareness level, which, in turn, depends on the behavior of multiple neighbors. The non-linearity of the dynamics of awareness and then of probabilities introduces a difference between the two types of agents that simulates the different impact on the epidemic prevalence of two subgroups of the population. Another difference between *Low* and *High Self-awareness* agents is that, by design, only *Low Self-awareness*

agents could show *negative awareness variations*. Again, this difference is useful to simulate subgroups with adaptive behavior, in this case a group that could intentionally seek to become infected.

Motivations for assuming different levels of awareness in a population, before an epidemic takes place, could be found in studies that have documented that, whatever the nature of the epidemic, there are individuals better equipped than others to face to it [34, 42, 43]. Education level, expertise, and in general the quality of knowledge an individual possesses are key for adopting effective countermeasures in mitigating the contagion risk [44–46], whereas the lack of knowledge or ingenuity may induce individuals to incautious behaviors, erroneous evaluations, and even to voluntarily engage in contagion spreading, as was recognized a long time ago for rumor and propaganda [47]. By introducing agent types, we aimed at capturing this fundamental aspect and use it for the analyses.

We combined these three types of agents to simulate how the infection prevalence could be modified by changing the proportion of one type with respect to another, as a possible effect of targeted awareness campaigns. The value of the *Self-awareness* factor is defined for each agent at setup and represents the fixed contribution to awareness. The following two factors are the variable part of awareness, dynamically depending to the spread of the contagion on the network.

**Imitation.**    It represents the main social reinforcement mechanism of our model, and depends on the state (Infected or not) of the neighbors. We assume that the imitative behavior is based on direct observation, therefore neighbors are assumed to take no active action. Depending on the combined effect of observations, which is governed by a threshold, an agent may or may not adapt its behavior. Agents type and the nature of the epidemic could determine either positive or negative variations of awareness, and as a consequence an agent, during a simulation, could become less likely to be infected or more likely (similarly for recovering).

**Communication.**    It represents a special form of communication, where an infected agent actively seeks to spread the contagion, more likely social contagion, to the neighbors. In this work, we have assumed that *direct messages* are the means of communication, and the effect on awareness is governed by a threshold. Similarly to the case of Imitation, it is the combined effect of infected peers to determine the possible behavioral response of an agent (i.e., change of awareness, followed by a possible change of state transition probabilities).

## Imitative behavior, intentionality, and beliefs

We add here some explanations and contextual descriptions with respect to the previous definition of factors contributing to agent awareness of our contagion-behavior model.

**Positive or negative effects of imitation.**    Imitation is a powerful mechanism that drives human behavior to adapt to a social context and often a safe strategy for decision-making in uncertain situations. The importance and prevalence of imitation as a key factor in determining individual beliefs and choices has been recognized in many studies [20, 48–51]. In our model, it is the process and mechanism that let agents' modify their awareness during the system evolution and from this to have a *complex contagion* process. Imitation, as a social enforcement, could be modeled in a variety of ways (e.g., based on equal or weighted relevance of the neighbors, dependent on peers more than 1-step away, or induced by broadcasting media), but it could also depend on the type of behavior that imitation induces, say a *positive* or a *negative* behavior. For sake of clarity, we do not assume any moral principle for defining what is a positive or negative behavior, just a generic utilitarian approach considering the social welfare. In particular, we assume that positive behaviors are those that more likely permit an agent to stay in the Susceptible state or to move from Infected to Recovered. Negative behaviors are those that more likely bring an agent to the Infected state. Accordingly, it is labeled as positive

a behavior that reduces the odds to get a disease, to start an addiction, or to believe in rumors; vice versa it is a negative behavior to increase them.

**Spreaders and messages.** A special form of social contagion, which is mostly related to the spread of rumors or false beliefs, is represented by an agent in the Infected state that believes in a certain rumor (e.g., false news, pseudo-scientific theory, conspiracy theory, and so forth) and *intentionally tries to spread it* by actively communicating with neighbors. For rumor epidemics, it has often been adopted a specific terminology for the states, like *Ignorant/ Spreader/Stifler* in place of Susceptible/Infected/Recovered [11]. We acknowledge the better fit of the specific terminology to the rumor case, but in this work we have preferred to maintain the traditional one for sake of homogeneity in presenting the different categories. We believe that no loss of precision or clarity is due to this choice. Research about learning or marketing has often debated how the *repetition of a message* is crucial in forming a belief. The fundamental reason is that in learning, opinion formation, and social media communication, there is typically a cognitive threshold, represented by a number of repetitions with the same content, above which an information is recognized as such (possibly unconsciously) and may contribute in forming knowledge (belief, opinion, preference) [52–54].

For clarity, it is worth noting that the case of rumor spreading is extremely rich in variations and scenarios, hence for the aim of this work, several simplifying assumptions were needed to analyze how awareness could play an important role within in the contagion-behavior model. Our first assumption is that the vector of rumor spreading is only represented by messages sent from an Infected node (a spreader) to its direct neighbors. No broadcast communication has been considered as well as the possibility of message forwarding. A second assumption is that spreading rumors is the consequence of a belief and agents could possibly change their mind, so they could "recover" from spreading or they could be "infected" and start over again. We do not consider the case of agents that spread rumors for reasons such as in advertising through influencers or disinformation campaigns. Another assumption is that Infected agents send messages regardless of any external variable, such as the density of Infected agents or the time elapsed from the beginning of the epidemic, and with constant frequency. We have also limited our case study to messages aimed at spreading the contagion, not at mitigating it, therefore the effect would be a negative variation of awareness, the same mechanism we defined for imitation, but through a different vector and a different threshold. Similarly to the imitation case, by design, only *Low Self-awareness* agents may adapt their awareness due to the effect of messages received from spreaders.

**Epidemic categories and intentionality.** We have considered three categories of epidemics: *disease*, *addiction*, and *rumor*, as the reference scenarios for modeling the network dynamics in case of only positive or both positive and negative awareness variations due to imitation, and in case of messages from infected peers.

For biological diseases, we assume that there is no intentionality in becoming infected, only in trying to avoid it (e.g., with medical care, healthy habits), whereas for addiction and rumor, we assume that all changes of state, e.g. becoming addicted or believing in false news, as well as recovering from them, are, at least partially, intentional. As a consequence, in case of disease epidemic, imitation can only have positive effects or otherwise no effect. We exclude the possibility of negative effects (e.g., one that voluntarily act to become infected). When the number of infected neighbors in a certain time frame exceeds a given threshold, then we assume that the awareness *Imitation* factor increases, and so does the total amount. For addiction, instead, the dynamic is more complex. An agent in Susceptible state, by observing its neighbors, may imitate the positive behavior (e.g., to stay away from addictions or do not believe in rumors), therefore *increasing* the value of the *Imitation* factor and the total awareness. But it may also decide to imitate the negative behavior (e.g., to start the addiction or to believe and spread

rumors). In this second case, the contribution of the *Imitation* factor is to *decrease* the total amount of awareness. Same logic applies for an agent in Infected state, it may decide to imitate Susceptible and Recovered neighbors and then to increase the chance to quit the addiction or to dismiss the false belief, or otherwise it may decide to imitate Infected neighbors and to further decrease the chance of rehabilitation or change of mind. The choice between positive or negative imitation is driven by a threshold on the number of Infected or non Infected neighbors. By changing the threshold, we could model agents with different attitudes, more or less likely to develop an addiction or spread rumors.

For the last category, rumor epidemic, we add the effect of messages sent from infected (spreaders) agents. By considering this category, our model of contagion-behavior accounts for the case of an agent in Susceptible state that could be influenced by the behavior of neighbors in two ways: *i)* by observing their state, as for typical imitative behavior, and ii) by the messages that a Infected neighbors send with a certain frequency, as a form of persuasion. With this scenario, it is possible to observe agents exhibiting a wide variety of probabilities to become infected or to recover, developing, at the end of simulations, a highly heterogeneous population with respect to initial equal state transition probabilities and (for agent type) awareness level.

Table 1 summarizes the characteristics of the three categories.

## Simple and complex contagion processes

With regard to the type of contagion process defined in our model, the base dynamics is the same for the three epidemic categories we have considered and it is a simple contagion: every contact between a Susceptible and an Infected agent has equal probability to trigger a change of state. However, state transition probabilities are specific of each agent and their values at each time step depend on the agent's social context. This introduces a form of *indirect* complex contagion dynamics, because multiple exposures to different neighbors are required to reach the imitation or message thresholds able to change the awareness through *Imitation* and *Communication* factors, which in turn may trigger a non-linear change in state transition probabilities. Therefore, with respect to the contagion type, our model is hybrid, with both simple and complex contagion effects: *i)* simple contagion as the simplified assumption for the base dynamics; and *ii)* complex contagion for changes of base rates (infection and recovery probabilities) driven by the corresponding adaptation, positive or negative, of awareness triggered by social reinforcement. To the best of our knowledge, this type of coevolving dynamics between epidemic rates and awareness is discussed here for the first time. A more advanced model would have included a complex contagion dynamics for contagion, too, this way combining two layers of social reinforcement: one for awareness dynamic adaptation and another for infection and recovery, possibly with different regimes (under some assumptions awareness dynamics could be regressive with respect to epidemic effects, in other cases it could be the opposite, with strong hype concerning the effects or extent of an epidemic). We guess that a study about the coevolution of these two dynamics, represented as both based on social reinforcement and, for the awareness, including both positive and negative effects could be a difficult but challenging research goal for future works.

## The model

We use a multiagent dynamic network model of N agents interacting within a network $\mathcal{N}$ [55]. The process, involving the agent network and driving agents time evolution, is an epidemic of a certain *category* (disease, addiction, or rumor) spreading on network $\mathcal{N}$. Agents are autonomous self-adaptive entities, each one characterized by a vector that, at each time step *t*, includes: the *state S/I/R*, *transition state probabilities*, *total awareness*, and *awareness*

**Table 1. Summary of differences between epidemic categories.**

| Category | Imitation | Intentionality |
|---|---|---|
| Disease | Positive Imitation | None |
| Addiction | Positive/Negative Imitation | Passive |
| Rumor | Positive/Negative Imitation and Messages | Active |

*contributing factors.* We now present the state transition diagram between S/I/R states and define transition probabilities as a function of awareness for each agent and at each time step. Next, we formalize our definition of awareness as a pooling function and give the second key definition of state transition probability as a function of an agent's awareness.

## State transition probabilities and awareness

With regard to probabilities in epidemic studies, it is common to focus on infection and recovery probabilities to describe a simple dynamic. Although it may look unusual, we choose instead to describe the contagion by referring to the *probability to remain susceptible* and *to recover*. The reason is because it permits more naturally a logical presentation of the effects of awareness variations. With this choice, it could be said that positive variations of agent awareness may produce increases in the two probabilities (i.e., to remain susceptible and to recover) and the effect is a reduction of the epidemic prevalence. Conversely, negative variations of awareness may decrease the two probabilities and the contagion tends to spread more. For the same reason, we preferred another notation for probabilities, in place of the Greek letters common in epidemic studies, to be more explicit about states and transitions. The notation is intuitive and has the form: $P_x$ for the probability to remain in state $x$, $P_{xy}$ for the probability to change state from $x$ to $y$. Therefore, probabilities are:

- $P_S$: probability to stay in Susceptible state, $P_{SI}$: infection probability (with $P_S = 1 - P_{SI}$);

- $P_{IR}$: probability to recover, $P_I$: probability to remain Infected (with $P_{IR} = 1 - P_I$);

- $P_{RS}$: probability to become Susceptible, $P_R$: probability to remain in Recovered state (with $P_{RS} = 1 - P_R$).

More formally, probabilities should show that they are specific of agent $i$ and depend on the awareness of $i$ at time step $t$. $P_{x,i}(Aw_{i,t})$ and $P_{x,y,i}(Aw_{i,t})$ are the more formal notations for $P_x$ and $P_{xy}$. For sake of simplicity, and when no ambiguity may arise, we will use the more compact form.

With respect to the relation between state transition probabilities and awareness, for $P_S$ and $P_{IR}$, we have assumed the S-shaped form of the *generalized logistic function* as a good approximation for that relation ($P_{\{S,IR\}} = logistic(Aw)$). Eq 1 shows the relation between $P_S$ and $P_{IR}$ with awareness, specific for each agent $i$ with a certain awareness at timestep $t$.

$$P_S(Aw) = P_S(0) + \frac{P_{max} - P_S(0)}{1 + m * \exp\left(\left[-k(Aw - Aw_{mid})\right]\right)^{\frac{1}{\nu}}}$$

$$P_{IR}(Aw) = P_{IR}(0) + \frac{P_{max} - P_{IR}(0)}{1 + m * \exp\left(\left[-k(Aw - Aw_{mid})\right]\right)^{\frac{1}{\nu}}}$$

(1)

In the equation, $Aw_{mid}$ defines the x-axes value of the sigmoid's midpoint, $k$ is the logistic growth rate, $m$ is related to the function sensitivity, and $\nu$ to the point of maximum growth. $P_S(0)$ and $P_{IR}(0)$ are the base rates, respectively, for $P_S$ and $P_{IR}$ when the agent has no awareness

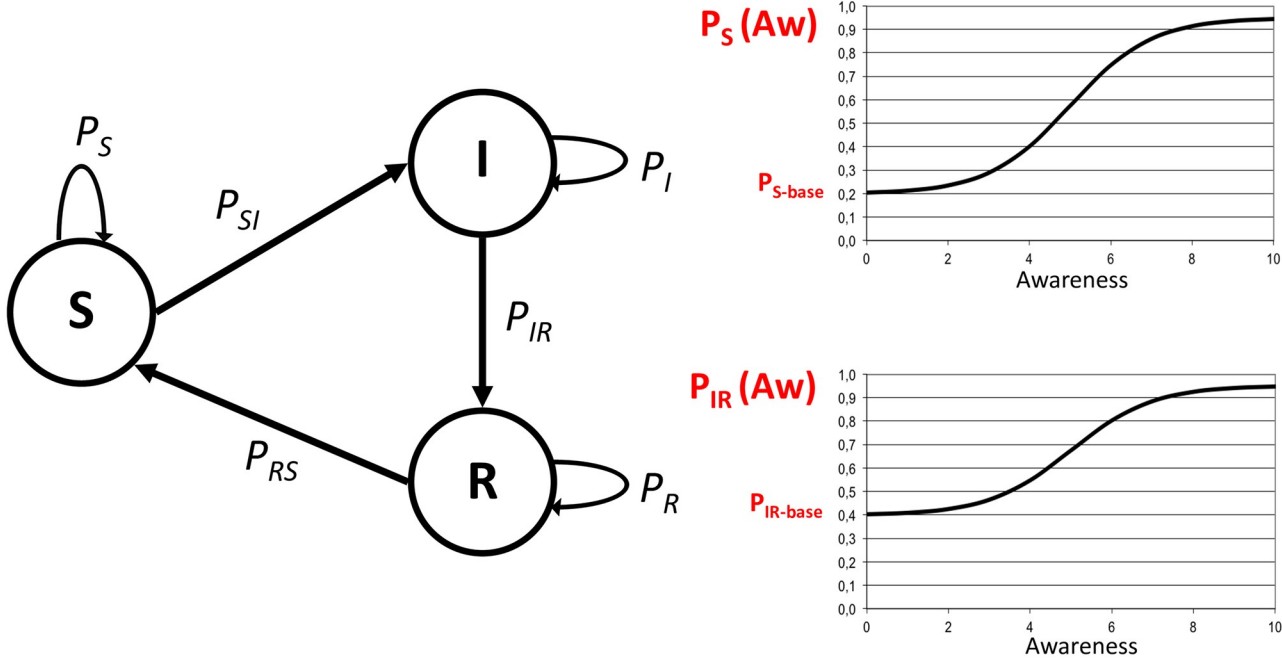

**Fig 1. States, probabilities, and awareness.** On the left, the state transition diagram for agent $i$ at time step $t$ is showed with corresponding probabilities. On the right, the logistic function $P(Aw)$ is showed in two examples for probabilities $P_S$ and $P_{IR}$, for agent $i$ at time step $t$. The maximum value of $x$-axis is $Aw_{max}$, the upper bound of awareness. On the $y$-axis, $Y(0)$ corresponds to base rates of transition probabilities, respectively $P_{S-base} = P_S(0) = 0.2$ and $P_{IR-base} = P_{IR}(0) = 0.4$, in the example.

$(P_S(0), P_{IR}(0) \neq 0)$, while it is possible to set an upper bound to the probabilities by means of $P_{max} \leq 1.0$.

Fig 1, on the right, shows two examples of probability as a function of awareness, $P_S(Aw)$ and $P_{IR}(Aw)$. Base rates are showed as $P_{S-base} = P_S(0)$ and $P_{IR-base} = P_{IR}(0)$.

The logistic function has been chosen because for small and for large values of $Aw$ the marginal gains are small, while in the middle range of $Aw$, probabilities are sensitive with respect to awareness variations. In practical terms, the S-shape means that for small values of $Aw$, it takes a certain amount of awareness variation to produce a change in agent behavior; for high values, instead, variations in awareness could only produce small effects having the agent already consistently changed its behavior, whereas in the middle range a variation of awareness may produce a sensible change in agent behavior. In general, this appears as a common way of reacting to reinforcements in social contexts. The typical reduced sensitivity of the logistic function for small and large values reinforces the complex contagion dynamics governing changes in probabilities. Not only for awareness variations a threshold should be met for imitation or messages, but especially for small values of awareness, the logistic function is insensitive to small changes, further stressing the need of a larger social reinforcement in order to change the contagion dynamics.

Up to now, we have ignored the probabilities $P_R$ and $P_{RS}$. The rationale is that the Recovered state and how long an agent remains in that state are associated to immunization, temporary or permanent. Immunization is key for diseases and a rich literature has modeled it in many ways. On the contrary, the concept of immunity is less clearly defined for addiction and rumor categories. Actually, there is no consistent notion of immunity for those categories, but only behavioral responses that may possibly be mapped on the Recovered state. For these reasons, in this work, we have simplified the model by assuming a fixed value for $P_R$ (smaller than

one, to have a SIRS model, instead of a SIR), and focusing our analyses on Susceptible and Infected states.

## Awareness factors

Our definition of awareness is that at time $t$ and for each agent $i$ of the network $\mathcal{N}$, awareness is defined as the geometric pooling of the three contributing factors here indicated as $\{a_1, a_2, a_3\}$, respectively with weights $\{w_1, w_2, w_3\}$ (i.e., $a_1 :=$ *Self-awareness*, $a_2 :=$ *Imitation*, and $a_3 :=$ *Communication*):

$$\text{Aw}_i(t) = \frac{a_{1,i}(t)^{w_1} a_{2,i}(t)^{w_1} a_{3,i}(t)^{w_1}}{Aw_{max}} \quad \text{with} \sum_{k=1}^{6} w_i = 1 \tag{2}$$

We have made some assumptions for the definition of $Aw_i(t)$. The first is that the maximum amount of awareness for an agent is bounded to an upper value $Aw_{max}$, which could serve as normalization factor. The meaning of this assumption is of cognitive boundedness, a common in autonomous agent studies [56]. The second assumption is to consider the geometric pooling a reasonable representation for the combination of heterogeneous awareness factors. Pooling functions are typically used for aggregating probabilistic opinions expressed, for example, by a panel of experts [57]. Here, similarly to the belief model of [34] for vaccinating behavior, we assume that our problem of independent information sources evaluated by an individual and contributing to his/her awareness is an acceptable approximation of the pooling problem.

This definition of awareness is useful for studying our reference epidemic categories. For example, for disease and addiction, where the *Communication* factor is not present, it suffices to set $w_3 = 0$ and reconfigure $w_1$ and $w_2$ so that $w_1 + w_2 = 1$. Also, to simulate scenarios where the different contributions have different relevance, for example reducing that of *Self-awareness* in favor of the variable factors, the weights of the pooling function offer an easy mechanism. To summarize, the whole dynamics of our contagion-behavior model could be expressed by combining a simple stochastic contagion dynamic with the two key definitions of $P_S$ and $P_{IR}$ as functions of awareness $Aw(.)$, as for Eq 1, and of awareness $Aw(.)$ as a pooling function of the three contributing factors, as for Eq 2.

## Schema of the coevolution dynamics

In Fig 2, we present a schema of how the contagion and the behavior dynamics coevolve. For simplicity, we have assumed the case of disease epidemic, with only *Imitation* and positive awareness variations. At *time step 0* the network is set up, assigning agents types, as for the required configuration, a corresponding value to *Self-awareness* ($a_1$), and base rates of $P_S$ and $P_{IR}$. All nodes, except for the seed node, are in Susceptible state. All other parameters (e.g., weights, $P_R$, etc.) are part of the standard configuration. Then, behavior and contagion coevolve in the following way: At *time step 1*, for each node in random order, first the behavior dynamic is updated: It is checked whether or not the number of infected neighbors exceeds the imitation threshold. If that is the case, then factor $a_2$ is increased with a predefined gain and the awareness of the agent is updated with Eq 2. This is the first behavioral response to the social reinforcement, the agent becomes more aware. Then, probability $P_S$ (i.e., the agent is in Susceptible state) is updated with Eq 1 using the newly calculated value of awareness. The new value of $P_S$ makes the agent more likely to stay Susceptible than in the previous time step. Now the contagion behavior is updated: it is a simple contagion, therefore, being the agent in Susceptible state, the infection probability $P_{SI} = 1 - P_S$, smaller than in *time step 0*, is used for each infected neighbor to decide whether or not the agent should change state. For the seed node,

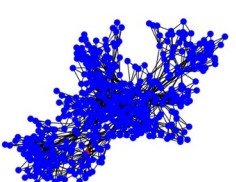
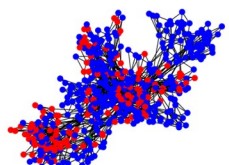

| time step 0 | time step 1 | time step 2 |
|---|---|---|
| **Set up:**<br>- N agents (mixed types)<br>- Network<br>- Base rate $P_S$, $P_{IR}$<br>- Num_infected := 1<br>- $Aw_i(0)= a_1(i)^{w1}/Aw_{max}$<br>- $Aw_{H\_type}(0) > Aw_{L\_type}(0)$ | For each agent $i$, randomly:<br>**Behavior dynamics:**<br>**Imitation (Positive)**<br>- infected neighbor threshold exceeded -> $a_2$++<br>- $Aw_i(1)= a_1(i)^{w1}a_2(i)^{w2}/Aw_{max}$<br>- $Aw(1) > Aw(0)$<br>- $(P_S, P_{IR})_i=logistic(Aw_i(1))$<br><br>**Contagion dynamics:**<br>- Seed node -> $P_{IR,i}$<br>- Others (for each infected neighbor)-> $P_{S,i}$ | For each agent, randomly:<br>**Behavior dynamics:**<br>**Imitation (Positive)**<br>*(same for timestep 1)*<br>- $Aw_i(2)$<br>- $(P_S, P_{IR})_i=logistic(Aw_i(2))$<br><br><br>**Contagion dynamics:**<br>- Infected state -> $P_{IR,i}$<br>- Susceptible state -> $P_{S,i}$<br>- Recovered state -> $P_{RS,i}$ |

**Fig 2. Schema of our contagion-behavior coevolution.** At *time step 0*, the set up configures the network, agent types, *Self-awareness* and probabilities. At *time step 1*, for each agent, first, the behavioral dynamics of awareness variations followed by probability variation takes place, then the stochastic contagion is carried out with probabilities possibly modified with respect to *time step 0*. *Time step 2* and the following ones proceed similarly.

being in Infected state, the imitation threshold will be checked with respect to the number of non-infected neighbors, awareness changed accordingly to Eq 2, and $P_{IR}$ updated with Eq 1. The recovery phase will use $P_R = 1 - P_{IR}$, this time just once, to decide if the state should change. Following time steps proceed in a similar way.

## Results

We have run simulations of different epidemic scenarios on an artificial network of $N = 10^3$ nodes for the analysis of aggregate metrics, and of $N = 10^4$ nodes to study the individual behavior of an agent population with heterogeneous features. The test network has been created to produce a clustering coefficient compatible with typical disassortative social networks [58] (i.e., approximately equals to 0.35), for which local cascade effects are common [59, 60]. In all simulations, a single seed node, selected at random, has been used. We are aware that it is not uncommon to identify multiple origins in epidemics and that therefore the single seed assumption may represent a strong simplification. We however believe, having done some tests, that considering multiple sources would not have substantially changed the significance of the results. On the other hand, a detailed analysis of the number and distribution of multiple origins would have changed the specific focus on awareness of this work.

Each data point presented in following figures has been averaged over, at least, 200 valid trials. Valid trials have been considered those that propagated the contagion at least for 5 time steps and at least infecting 50 nodes (5% of the total population size, N = 1000). These two conditions have been set empirically given the fact that: *i)* at *time step 3* most simulations are already close to the peak of infected nodes; *ii)* the total of 50 infected nodes is sufficient to discard the (rare) cases where the contagion seemed to jump back and forth for several time steps among few nodes. Given the reference configuration that we used throughout our simulations,

with an infection probability $P_{SI}$ = 0.2, a recovery probability $P_{IR}$ = 0.5, and the presence of only a single seed node, the large majority of non-valid trials corresponded to simulations that terminated within the first two time steps and with a number of infected nodes smaller than 50. For each unique configuration whose result are presented, the total number of simulations we run was between 800 and 1000, including valid and non-valid ones. We found a reference to confront our validity criteria. In [36], "an outbreak is defined as a minimum final epidemic size of 25 (i.e. 0.5% of the total population size N = 5000)" (page 3, caption of Fig 2). Infection and recovery probabilities are both equal to 0.1 in their stochastic simulations, and in total they run 10000 simulation for unique network. Given the different network sizes and probabilities, it seems to us that our criteria to consider a trial as valid is comparable to the one assumed in [36].

Simulation results are presented in the reminder of this section starting with some examples of network dynamics with simple contagion only (no awareness dynamics), then followed by results related to our reference epidemic categories. For each group of simulations, a table summarize the main parameters and their values. Table 2 shows the parameters with fixed values for all groups of experiments that follows. Values have been defined empirically, as a result of many trials, and selected for qualitative analyses of the results.

## Pure epidemic, no awareness

With these simulations, we aim at showing the basic behavior of our network for classical epidemic models and to set the benchmark for subsequent analyses when the effect of different components of awareness will be studied. Here awareness is not considered and state transition probabilities do not dynamically change.

**Table 2. Configuration values for different groups of simulations.** When not repeated, a parameter value is intended to be the same of the previous simulation category.

| Positive imitation | |
|---|---|
| **Parameter** | **Value** |
| $(P_S, P_{IR}, P_R)$ (base rates) | (0.8, 0.6, 0.5) |
| $T_{imitation}$: Imitation threshold (H and L agent types) | 0.5 (majority rule) |
| *H type agent*: *Self-awareness* ($a_1$) | 10.0 |
| *L type agent*: *Self-awareness* ($a_1$) | 1.0 |
| *H type agent*: *Imitation* ($a_2$) when $T_{imitation}$ exceeded | $a_2 + 3.0$ |
| *L type agent*: *Imitation* ($a_2$) when $T_{imitation}$ exceeded | $a_2 + 0.5$ |
| *Awareness weights*: ($w_1, w_2$) | (0.5, 0.5) |
| **Positive and Negative imitation** | |
| **Parameter** | **Value** |
| $T_{imitation}$: Imitation threshold (H types only) | 0.5 (majority rule) |
| *H type agent*: *Positive Imitation* ($a_2$) when $T_{imitation}$ exceeded | $a_2 + 3.0$ |
| $T_{imitation}$: Imitation threshold (L types only) | (0.2, 0.5, 0.8) |
| *L type agent*: *Positive Imitation* ($a_2$) when $T_{imitation}$ exceeded | $a_2 + 0.5$ |
| *L type agent*: *Negative Imitation* ($a_2$) when $T_{imitation}$ not exceeded | $a_2 := 0.2$ |
| **Messages, Positive and Negative imitation** | |
| **Parameter** | **Value** |
| $T_{messages}$: Number of received messages threshold (L agent types) | (8, 10) |
| *L type agent*: *Communication* ($a_3$) when $T_{messages}$ exceeded | $a_2 := 0.2$ |
| *Awareness weights*: ($w_1, w_2, w_3$) | (0.33, 0.33, 0.33) |

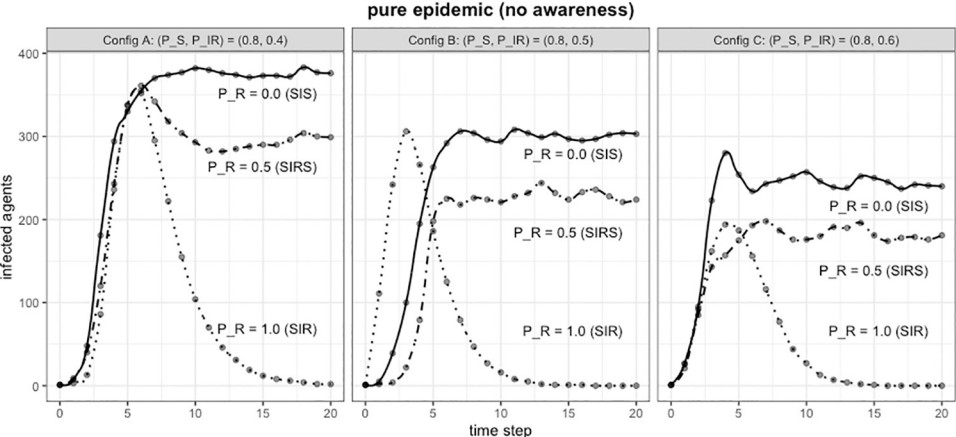

**Fig 3. Pure epidemics, no awareness: SIS/SIR/SIRS epidemic models for different configurations.**

Fig 3 shows the behavior of SIS/SIR/SIRS epidemics models on our test network for three configurations (i.e., A, B, and C). For each configuration, we set the probability to stay in the Susceptible state ($P_S$ = 0.8) and correspondingly the probability to get infected ($P_{SI} = (1 − P_S) = 0.2$). By increasing the probability to recover ($P_{IR}$ = (0.4, 0.5, 0.6)), the maximum number of infected decreases of about 20%. Instead, by changing the probability to stay in the Recovered state ($P_R$ = (0.0, 0.5, 1.0)), the classical SIS/SIR/SIRS epidemic models could be obtained.

### Positive imitation

With these simulations, we start testing the effect of awareness by considering two components, namely the *Self-awareness* and the *Imitation* factors (respectively, $a_1$ and $a_2$). Here only positive variations of the awareness are possible, according to the disease category.

**Configuration.** In this set of simulations, the difference between L and H type agents (defined at set up by the value of factor $a_1$) is that the former, individually, produce small improvements in state transition probabilities $P_S$ and $P_{IR}$, while the impact of the latter is larger. The effect is a consequence of the difference in $a_1$ initial values and the different gain of $a_2$. No other difference in action rule is implemented. With respect to a realistic scenario, we have assumed that L type agents could include a large proportion of the population (up to the entire population), while H type agents could only be a minority. The rationale for this assumption is based, for example, on the cost of awareness campaigns or the time needed to improve individual awareness up to high levels.

**Simulation results.** In Fig 4, results are obtained, left to right, starting from a population of only *No awareness*, therefore only simple contagion dynamics, then with mixed populations of L and H types, first the proportion is 90/10, then 70/30. With these experiments we want to show the variability of results and how the awareness dynamics tends to reduce the prevalence of infection and the duration of an epidemics.

Fig 5 gives the full pictures of all experiments with heterogeneous population and positive awareness variations. Experiments are conducted starting with a population of *No awareness* agents, then introducing an increasing proportion of *L type* agents, until a full population of L types is reached. At that point, the role of H type agents is evaluated, by introducing them in different proportions. The rational behind these simulations is, first, to study how a small degree of awareness distributed over a large fraction of (or the entire) population is able to

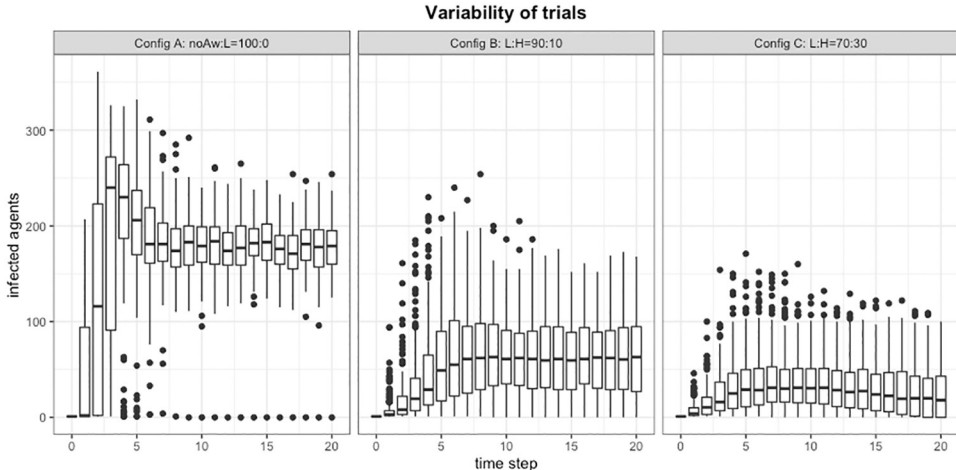

**Fig 4. Variability of the contagion dynamics.** The results have been obtained based on 200 valid trials for time steps. It should be noted, from left right, the increasing presence of trials with a number of infected nodes that dropped to zero. One of the effect of the awareness dynamics was to reduce the prevalence of infected but even terminate more often the epidemics. The large variability of initial steps should be also remarked. *Left*: the baseline case of pure epidemics with no awareness mechanism; *Center*: the case of positive imitation with proportion of L/H type equals to 90:10; *Right*: positive imitation with L/H type proportion equals to 70:30.

change the epidemic dynamics. Then to study how, by empowering a subset of agents with a higher degree of awareness, the dynamics could be further modified.

In Fig 5, it could be observed the clear effect of awareness dynamics on the number of infected and on the overall process. The progressive reduction of infected clearly emerges soon when some *No awareness* agents are converted in L types. With a proportion 70/30 of NoAw:L types, the reduction is already larger than 10%. However, the most striking result is when the

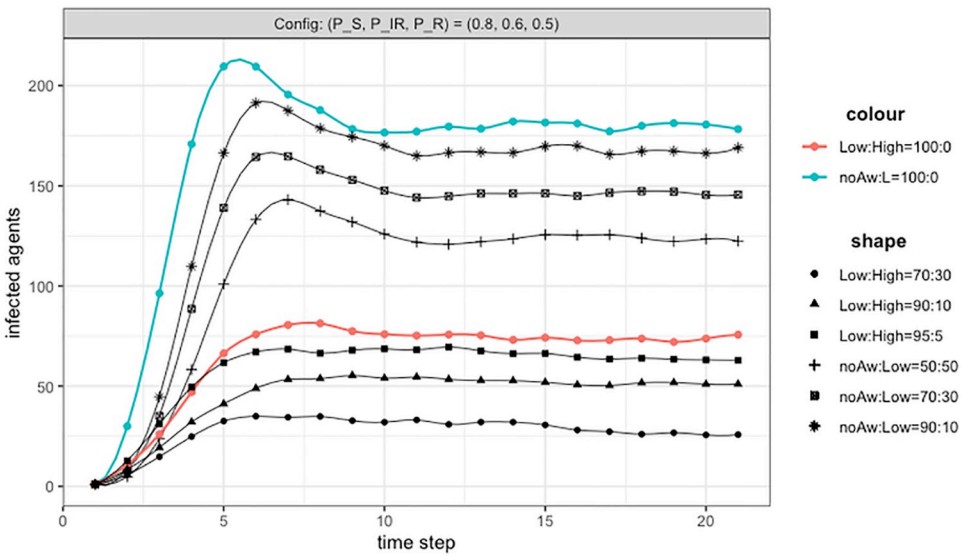

**Fig 5. Positive imitation: Effect of awareness and agent type.** Threshold for imitation is 0.5 (majority rule). Colored lines are for the two cases of uniform population: all *No awareness* agents (blue line) and all *L types* (red line). Lines with different shapes are for mixed populations: no awareness/L type and L/H types. The results presented here have been produced by running more than 6000 simulations, to obtain at least 200 valid trials for each configuration.

whole population is of L types. At that level, the difference with the single contagion is large (>50%) and the dynamics has changed, apparently mitigating the initial exponential increase. The following introduction of H types further improves the results, although not dramatically.

These results, although qualitative and not generalizable out of our artificial network, could nevertheless provide some useful advices. First, they remind of the power of a small but largely spread intervention to raise awareness, which could have a surprisingly positive impact on a population. An objection is that a small increase in awareness may have immediate effects, but it easily disappear if not sustained in the long run. Another advice is about the importance of mitigating the dynamics, not just the number of infected. A contagion with an exponential initial dynamics not just may reach a dangerous peak of infected, but it exhibits a wild initial variability that could make the response to an epidemics extremely difficult. The unpredictable behavior of early stages of an epidemic, easily dominated by stochastic uncertainty or subtle effects of heterogeneity and asymmetries, could induce severe evaluation errors when used to forecast future evolution. Unfortunately, in many real situations and for epidemics of different nature, it is in the early phases that the pressure of emotional reactions, for example by the media or politicians, may lead to decisions producing over- or underreactions.

By changing the the imitation threshold, populations of agents more or less like to take precautions when infected peers are observed could be simulated. For other parameters, tuning the gain of factor $a_2$ of awareness for L and H types (now with a proportion of 1:10), different scenarios could be represented.

A different viewpoint is presented in Fig 6. Probability $P_S$ and $P_{IR}$ are adjusted according to the logistic function based on the varying degree of awareness gained by agents during the temporal evolution of the epidemic. Fig 6 shows, for each agent (the $x$ axes lists the IDs from 1 to 10000), the value of $P_S$ (*top*) and $P_{IR}$ (*bottom*) at the end of the simulation (time step 20). The difference between L and H types is clearly visible for both probabilities, as well as a different variance that the two types develop starting from the same base rates ($P_{S-base}$ = 0.8 and $P_{IR-base}$ = 0.6). From this figures, the stochastic heterogeneity introduced by our awareness mechanism into state transition probabilities appears in a clear way, and represents, in our opinion, a more realistic representation of a population of individuals for epidemic models and spreading phenomena.

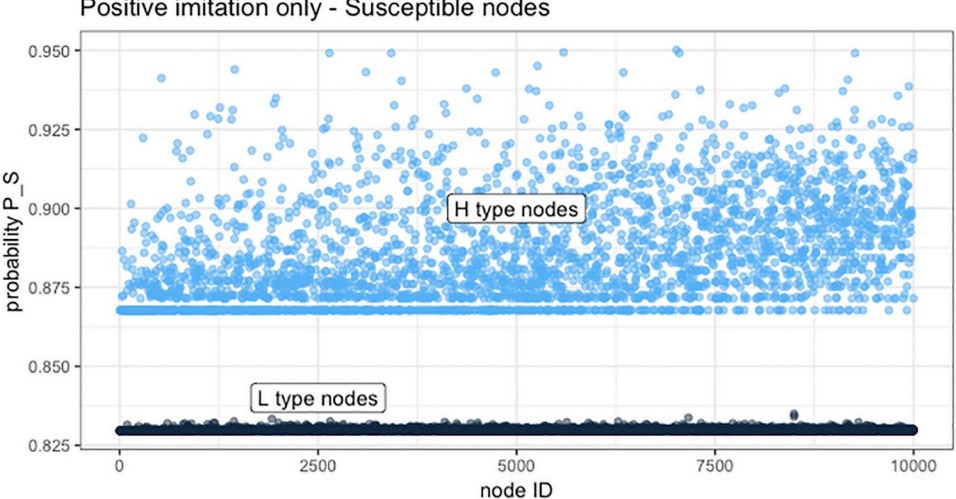

**Fig 6. Positive imitation: Distribution of probability values among L and H agent types.** On the $x$ axes, agent IDs in increasing order; on the $y$ axes, probability $P_S$ for Susceptible agents (*top*) and $P_{IR}$ for Infected agents (*bottom*) in a sample trial.

## Positive and negative imitation

We move now to the next set of simulations, with a more complete characterization of imitation. Different from the previous case, which essentially modeled a disease epidemic, here we consider the case of addictions and the spreading of rumors.

**Configuration.** An individual in addiction or rumor epidemics may act by imitating others in two possible ways: by being nudged to adopt a *positive behavior*, meaning that it results in an increase of $P_S$ and $P_{IR}$, or to adopt a *negative behavior*, meaning that the outcome will be a decrease of $P_S$ and $P_{IR}$. We have called *Positive Imitation* the first case and *Negative Imitation* the second one.

Table 2 shows the different parameters, with respect to the previous case. The main difference is that H type agents and L types have distinct imitation thresholds for the two different behaviors. We tested different thresholds for Negative Imitation, from 0.8 to 0.2, meaning from L type agents that very likely will behave negatively, to the opposite case.

Therefore, to summarize, Positive and Negative Imitation have the following meaning and targets:

- *H type agent—Positive Imitation*: Same as disease epidemic, but limited to H type agents. A Susceptible H type agent becomes more aware of the epidemic and thus adopts countermeasures when the number of Infected nodes directly connected (addicted or rumor spreading) exceeds a certain threshold. With Positive Imitation, *awareness increases* and, accordingly, $P_S$ tends to increase. Similarly if the H type agent is Infected, its increased awareness tends to speed up the recovery by increasing $P_{IR}$.

- *L type agent—Positive and Negative Imitation*: L type agents now are assumed to behave as pure imitators. Therefore, after defining a target *threshold* of directly connected agents, if the proportion of directly connected peers in Infected state exceeds the threshold, then a L type agent will seek to increase its odds to become Infected by *reducing awareness* that, in turn, will reduce $P_S$ or $P_{IR}$. Vice versa, if the proportion of directly connected peers in Infected state does not exceed the threshold, the L type agent will imitate the Susceptible or Recovered by increasing awareness and then increasing $P_S$ or $P_{IR}$.

**Simulation results.** Fig 7 shows an example of the effect of Negative Imitation on agents that, during a simulation, had mixed Positive and Negative Imitation (at certain time steps they behaved as Positive Imitation, in others as Negative Imitation) (*blue dots*), with respect to agents that experienced only Positive Imitation (*red dots*). In the model, Negative Imitation is produced by setting the *Imitation* factor *a2* to a value between (0,1), which reduces the total awareness, when geometric pooling defined in Eq 2 is calculated. Fig 7 represents the value of $P_{IR}$ at the end of the simulation (N = 10000). In this simulation, Negative Imitation is triggered according to majority rule (threshold = 0.5), meaning that an L type agent imitates the behavior of the majority of its neighbors. The overall effect of introducing Negative Imitation is to have a larger distribution of $P_{IR}$ values with respect to the case of only Positive Imitation, with a richer dynamics during a simulation.

In Fig 8, simulations have been run by changing the threshold for triggering the Negative Imitation of L type agents (the values tested for the threshold are 0.2, 0.5, and 0.8). The overall effect of a larger proportion of H types is more evident than what we have seen in previous Fig 5, because here the benefit of the L type agents is mitigated by those that negative imitate. Confronting the values between Figs 8 and 5 for threshold = 0.5, it can be seen the effect of negative imitation in the higher prevalence of infected at the end of the simulation, in case of negative imitation.

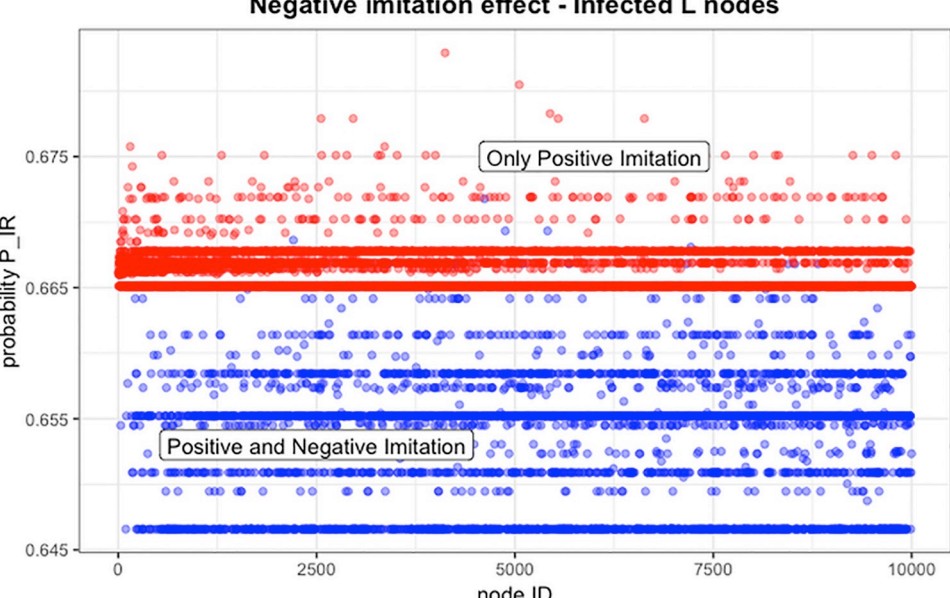

**Fig 7. Positive and negative imitation: Distribution of probability values for L type agents.** On the *x* axes, agent IDs in increasing order; on the *y* axes, probability $P_{IR}$ for Infected agents in a sample trial. (*blue dots*): agents experiencing both Positive and Negative Imitation; (*red dots*): agents experiencing Positive Imitation only.

From these results, we have seen how the interplay between the outcome of an epidemic, when a degree of intentionality in individual behaviors is considered, could be described with a rich set of features. Individuals may react differently to external stimuli according to different levels of awareness (e.g., by elaborating a strategy for reducing the risk or according to a herding behavior [61]). Some of them may even react incoherently when the whole epidemic event

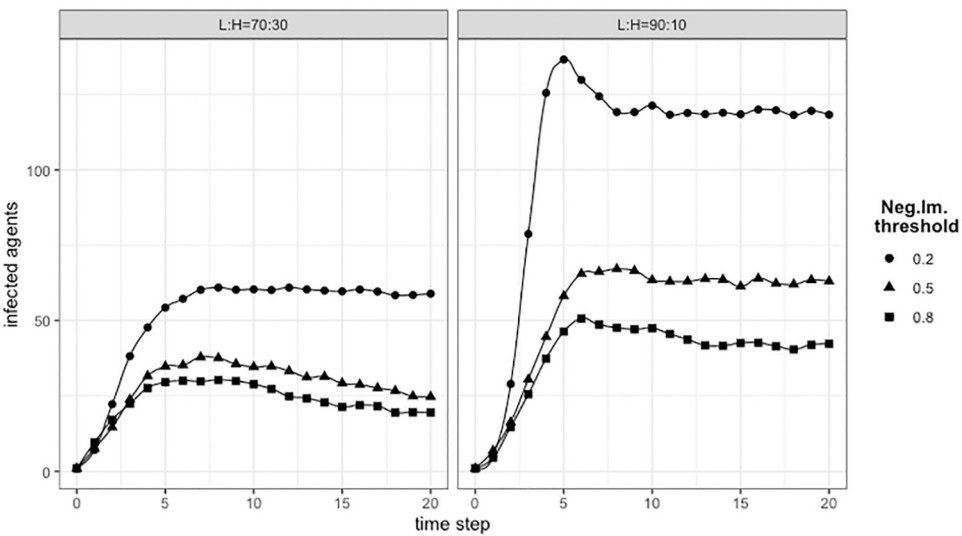

**Fig 8. Positive and negative imitation: Infected with different thresholds.** Threshold equals 0.8 means that an L type agent imitates Infected (Negative Imitation) when 20% of its neighbors are Infected and imitates not Infected (Positive Imitation) when 80% of neighbors are not Infected. For thresholds 0.5 and 0.2, Negative Imitation is triggered when the proportion of Infected neighbors is, respectively, 50% and 80%.

is considered (e.g., sometimes being influenced by a majority of peers with a certain behavior, other times by a different behavior model), and the individual attitudes towards risk of contagion influenced by a variable awareness, here represented by state transition probabilities, might distribute within a range of values. We stress the importance of distinguishing epidemic categories based on intentionality of individual actions, because, even for models relatively simple as the one we have studied, the nature of some fundamental interactions should be revised, with respect to the traditional approach to epidemic models based on diseases.

## Messages

In this last section of the experimental results, we have added messages sent by Infected agents to the neighbors. Here the mechanism is presented in a basic form, but nevertheless relevant to better specify another possible difference between our epidemic categories.

**Configuration.** Again, Table 2 shows the different parameters, with respect to the previous case. In this case, the difference is the introduction of a new threshold ($T_{m}essages$) to match with the number of messages received by an agent in Susceptible state from Infected nodes. Factor *Communication* ($a_3$) keeps the messages count. This represents a form of *dose-based diffusion*, with a memory effect that disappears only once the threshold is exceeded. When the threshold ($T_{m}essages$) is exceeded, an agent decreases the awareness. For simplicity we used the same mechanism of Negative Imitation, reducing the value of $a_2$. At that point, the queue of messages is also emptied, therefore there is no accumulation, or memory effect, once a behavioral response has been produced. It is a simplified assumption to avoid an excessive prevalence of the effect. In our simplified set up, messages are sent at constant rate by Infected agents to neighbors as an attempt to recruit more spreaders. In particular, this means that messages have the only possible effect to increase the odds of a state transition from Susceptible to Infected.

**Simulation results.** Fig 9 shows the experimental results for two agent populations with different proportion of L/H agents. Here the dynamics combines four different behaviors: H type agents positive imitating, L types positive and negative imitating, and L types message-driven negative imitation. For imitation we used $T_{imitation} := 0.5$ to compare the results with Figs 5 and 6. The colored lines indicated as "no Imit, no Msg" and "Imit, no Msg" corresponds, respectively, to (NoAw,L) := (100,L) of Fig 5, the case of single contagion, and to (T := 0.5) of Fig 6, the positive and negative imitation with majority rule.

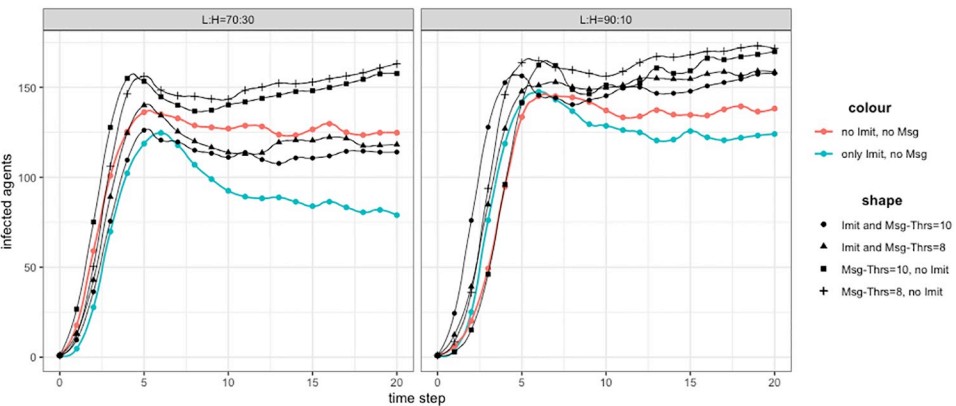

**Fig 9. Messages and imitation: Effect of different configurations in two populations.** On the *x* axes, simulation time step; on the *y* axes the number of Infected (Spreader) agents. (*left*): L:H = 70:30; (*right*): L:H = 90:10. Colored lines are for setup with only Imitation mechanism and no Messages, and neither Messages nor Imitation (pure epidemic). Lines with different shapes are for configurations with Message threshold.

The results, as expected, show an increased number of infected, with respect to previous experiments. For the *L:H = 70:30* configuration, it can be clearly observed the negative effect of messages increasing the number of Infected agents. For the *L:H = 90:10* configuration, the result is qualitatively similar but with a difference worth to be noted. The difference between the two benchmarks (lines red and blue) is reduced due to the smaller number of H agents contributing to Positive Imitation. This makes the negative contribution of messages proportionally more relevant, with the consequence that all lines with different shapes, representing configurations with message thresholds, exceed the number of Infected of both benchmarks. Again, presenting these results, we wish to highlight the complex interplay of the different contributions to awareness resulting in different types of behavior. Finally, as expected, configurations with smaller message thresholds ($Msg_{Thrs}$ = 8 with respect to $Msg_{Thrs}$ = 10) produce proportionally more Infected nodes.

## Conclusion

Previous epidemic models have typically included individual awareness as a factor for varying the state transmission probabilities. We made a step further by modeling awareness as a combination of contributing factors. This has let us introduce the main contribution of this work, that is the modeling of positive and negative variations of awareness. This, combined with heterogeneous agent types, different for level of adaptation of awareness and behavior, represent our contagion-behavior model. The results have showed how both the number of infected and the dynamic of the epidemic could be strongly modified. With positive imitation we have showed strong changes also in the initial phase, exponential and with extreme variability in a typical stochastic contagion. With negative imitation and negative effects driven by messages, instead, we have showed that, with a richer setting, the results may change, sometimes even with performances worse than simple contagion. Therefore, it is a rich picture what it emerged from our experiments. The model presented and the experimental results could be useful for future studies about the coevolution between individual awareness and spreading phenomena. The two dynamics are entangled in all epidemics at different degrees, and its modeling is critical for the definition of control measures aiming at mitigating negative effects (or at amplifying positive effects of beneficial spreading phenomena). The effect of media coverage and of education campaigns are tightly dependent on the impact on individual awareness and how it influences behaviors. From the spread of diseases, addictions, to the many facets of rumor (e.g., ideas, memes, fashions, popular beliefs), and all relevant network epidemic categories that we have not considered in this work, the ability to simulate and analyze the dynamics between a contagion, social or biological, and the awareness of people could be important in several situations. Policy makers and educators, for instance, often address awareness campaigns to groups of people at risk (e.g., typically selected for a combination of age, gender, ethnicity, income, education level, habits, health status, etc.). The increase of individual awareness may result in different outcomes: on the positive side, the individual less likely will turn infected (addicted or prey of false beliefs) and (s)he might have a beneficial network effect on peers through social learning and imitation; on the negative side, it is also possible that the individual will quickly lose the awareness gained, when exposed to negative social reinforcements. The outcome of a real awareness campaign is most often a combination of all these outcomes, and a model should account for all of them. Another application of the model could be in supporting the evaluation of an awareness campaign with different targets. An important decision for a campaign planner could be to choose between reaching a large audience with no awareness with the goal to raise it to a low but not negligible level, or to focus on a smaller group and make them well aware. This is an instance of the classical moral dilemma of how to

distribute scarce resources: providing a few to many or a lot to a few? For an awareness campaign planner it represents a hard decision, for the ethical questions and the utilitarian considerations it raises. These considerations about the model applicability and a paper recently appeared [62] have inspired what, in our opinion, could be an interesting development: *group awareness* and *group contagion/behavior dynamics*, that would introduce a higher order model of complex contagion and awareness dynamics, even more representative of social contexts characterized by local effects, homogeneous clusters, and small cascades [63]. Finally, our contagion-behavior model, while it proposes some original solutions, it also comes with many limitations. One is the simplification related to the simple contagion dynamic. It should be removed and the model should fully become a complex contagion dynamics. Then there are the many limitations and assumption of our mechanisms. The case of rumor epidemics and messages is an extreme simplification of the richness of the spread of rumors, ideas, false information, propaganda, etc. We have barely scratched the surface of those models. The study of positive and negative awareness variations is just at the beginning, there could be countless nuances and interesting scenarios to explore. We have only sketched the basic mechanism. The static contact network is another simplification that we accepted, but for which we are well aware that it has to be changed. Networks should be dynamic, as well as populations. With respect population, we wanted to introduce heterogeneity, as we did, but clearly our agent types are a vague approximation of network communities and profiles. These are some of the many limitations of our work. We hope that they are regarded as necessary simplifications for an initial study that allows for the simulation of a rich contagion-behavior dynamics and considers awareness as a dynamic property of individuals, coevolving in a complex contagion process.

## Author Contributions

**Conceptualization:** Marco Cremonini.

**Data curation:** Samira Maghool, Marco Cremonini.

**Formal analysis:** Nahid Maleki-Jirsaraei, Marco Cremonini.

**Investigation:** Marco Cremonini.

**Methodology:** Marco Cremonini.

**Project administration:** Marco Cremonini.

**Resources:** Nahid Maleki-Jirsaraei.

**Software:** Samira Maghool, Marco Cremonini.

**Supervision:** Marco Cremonini.

**Validation:** Samira Maghool, Marco Cremonini.

**Visualization:** Marco Cremonini.

**Writing – original draft:** Marco Cremonini.

**Writing – review & editing:** Samira Maghool, Nahid Maleki-Jirsaraei, Marco Cremonini.

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
