## [Decision Letter · Decision Letter 0]

21 Aug 2019

PONE-D-19-20285

A multicomponent model of awareness for different categories of network epidemics

PLOS ONE

Dear Dr. Cremonini,

Thank you for submitting your manuscript to PLOS ONE. After careful consideration, we feel that it has merit but does not fully meet PLOS ONE’s publication criteria as it currently stands. Therefore, we invite you to submit a revised version of the manuscript that addresses the points raised during the review process.

We would appreciate receiving your revised manuscript by Oct 05 2019 11:59PM. To enhance the reproducibility of your results, we recommend that if applicable you deposit your laboratory protocols in protocols.io, where a protocol can be assigned its own identifier (DOI) such that it can be cited independently in the future. For instructions see: http://journals.plos.org/plosone/s/submission-guidelines#loc-laboratory-protocols

We look forward to receiving your revised manuscript.

Kind regards,

Lidia Adriana Braunstein, Phd in Physics

Academic Editor

PLOS ONE

Journal Requirements:

2. Our internal editors have looked over your manuscript and determined that it is within the scope of our Science of Stories Call for Papers. This collection of papers is headed by a team of Guest Editors for PLOS ONE: Peter Sheridan Dodds, Mirta Galesic, Mohit Iyyer, and Matthew Jockers. Additional information on this Call for Papers can be found on our website: https://collections.plos.org/s/science-of-stories. Please note: Sharing the data underlying the findings will be a requirement of publication, per the PLOS data policy. As applicable, under this Call for Papers, authors are expected to provide, upon submission, the source code needed to replicate their findings, ideally in a repository (such as Zenodo, which can also import from GitHub) or a suitable cloud computing service (such as Code Ocean). Please specify whether or not you would like your manuscript to be considered for the collection in your cover letter.

3. In order to meet the requirements for the Science of Stories collection, the Guest Editors ask that you please make the code to reproduce your analysis available in a stable, public repository (for example, Zenodo, or GitHub) or a suitable cloud computing service (such as Code Ocean) when submitting your revised manuscript. The code should include a license file and detailed readme so that someone with access to the dataset is able to reproduce your analysis using the code. We ask that you include the DOI for the repository holding your code in an updated Data Availability statement with your revised manuscript.

Reviewers' comments:

Reviewer's Responses to Questions

**Comments to the Author**

1. Is the manuscript technically sound, and do the data support the conclusions?

Reviewer #1: Yes

Reviewer #2: No

2. Has the statistical analysis been performed appropriately and rigorously? 

Reviewer #1: Yes

Reviewer #2: No

3. Have the authors made all data underlying the findings in their manuscript fully available?

Reviewer #1: Yes

Reviewer #2: No

4. Is the manuscript presented in an intelligible fashion and written in standard English?

Reviewer #1: Yes

Reviewer #2: No

5. Review Comments to the Author

Reviewer #1: Comments on ‘A multicomponent model of awareness for different categories of network epidemics’

The coevolution spreading of epidemic and awareness on networks is a hot topic in recent years. In this paper, the authors studied the epidemic spreading with awareness, and found some interesting phenomena. I want to recommend its publication in PLoS ONE. I have a few minor comments.

1. The coevolution of epidemic and awareness is a hot topic and closely related to this work. The author should discuss more, and some related papers are https://doi.org/10.1016/j.physrep.2019.07.001; Scientific Reports 6 (2016) 29259; Scientific Reports 4 (2014) 5097.

2. I think the evolution mechanisms of the three different dynamics disease, addiction, and rumors are similar, that is they are simple contagions. Once the contagion with social reinforcement, I think the situation is different. Therefore, the authors should be specific that their research object is simple contagions and discussed what will happen for complex contagions in the discussion section.

3. The model description is not very clear, and please improve it.

4. The resolution of the figures is meager, and please improve them.

5. Add some applications about the results.

Reviewer #2: The paper entitled “A multicomponent model of awareness for different categories of network epidemics” studied the effect of individual awareness which is a function of several different contribution factors pooled together. And this study may help to understand the key feature in the epidemic spreading. However, due to the numerous deficiencies in this paper, I decide not to recommend this manuscript to publish in PONE. And the main issues raised by me are as follows:

1. The figures are too messy and too blurred.

2. Too many experimental parameters are not given.

3. The authors’ workload is insufficient, for example, too few experiments were repeated which bring about the messy lines in the figures.

4. In the section multicomponent awareness, the authors described a lot of useless content.

5. The logic between experiment results and the conclusions is not strong.

6. PLOS authors have the option to publish the peer review history of their article (what does this mean?). If published, this will include your full peer review and any attached files.

Reviewer #1: No

Reviewer #2: No

---

## [Author Response · Author response to Decision Letter 0]

8 Oct 2019

We have responded to all Reviewers comments in the Response to Reviewers, which we have included in this submission. The manuscript has been marked in the modified parts with references to Reviewers comments.

---

## [Decision Letter · Decision Letter 1]

6 Nov 2019

The coevolution of contagion and behavior with increasing and decreasing awareness

PONE-D-19-20285R1

Dear Dr. Cremonini,

We are pleased to inform you that your manuscript has been judged scientifically suitable for publication and will be formally accepted for publication once it complies with all outstanding technical requirements.

With kind regards,

Lidia Adriana Braunstein, Phd in Physics

Academic Editor

PLOS ONE

Additional Editor Comments (optional):

Reviewers' comments:

Reviewer's Responses to Questions

**Comments to the Author**

1. If the authors have adequately addressed your comments raised in a previous round of review and you feel that this manuscript is now acceptable for publication, you may indicate that here to bypass the “Comments to the Author” section, enter your conflict of interest statement in the “Confidential to Editor” section, and submit your "Accept" recommendation.

Reviewer #1: All comments have been addressed

Reviewer #2: All comments have been addressed

2. Is the manuscript technically sound, and do the data support the conclusions?

Reviewer #1: Yes

Reviewer #2: Yes

3. Has the statistical analysis been performed appropriately and rigorously? 

Reviewer #1: Yes

Reviewer #2: Yes

4. Have the authors made all data underlying the findings in their manuscript fully available?

Reviewer #1: Yes

Reviewer #2: Yes

5. Is the manuscript presented in an intelligible fashion and written in standard English?

Reviewer #1: Yes

Reviewer #2: Yes

6. Review Comments to the Author

Reviewer #1: The authors have addressed all my comments. I want to recommend it’s publication in plos one. Publish as it is

Reviewer #2: (No Response)

7. PLOS authors have the option to publish the peer review history of their article (what does this mean?). If published, this will include your full peer review and any attached files.

Reviewer #1: No

Reviewer #2: No

---

## [Editor Report · Acceptance letter]

19 Nov 2019

PONE-D-19-20285R1 

The coevolution of contagion and behavior with increasing and decreasing awareness 

Dear Dr. Cremonini:

I am pleased to inform you that your manuscript has been deemed suitable for publication in PLOS ONE. Congratulations! Your manuscript is now with our production department. 

With kind regards,

on behalf of

Dr. Lidia Adriana Braunstein 

Academic Editor

PLOS ONE